# Genetic associations between circulating immune cells and periodontitis highlight the prospect of systemic immunoregulation in periodontal care

Xinjian Ye[1], Yijing Bai[2], Mengjun Li[1], Yuhang Ye[1], Yitong Chen[3], Bin Liu[4], Yuwei Dai[1], Shan Wang[1], Weiyi Pan[1], Zhiyong Wang[1], Yingying Mao[4]*, Qianming Chen[1]*

[1]Stomatology Hospital, School of Stomatology, Zhejiang University School of Medicine, Zhejiang Provincial Clinical Research Centre for Oral Diseases, Key Laboratory of Oral Biomedical Research of Zhejiang Province, Cancer Centre of Zhejiang University, Hangzhou, China; [2]The First School of Clinical Medicine, Zhejiang Chinese Medical University, Hangzhou, China; [3]School of Stomatology, Zhejiang Chinese Medical University, Hangzhou, China; [4]Department of Epidemiology, School of Public Health Zhejiang Chinese Medical University, Hangzhou, China

**\*For correspondence:**
myy@zcmu.edu.cn (YM);
qmchen@zju.edu.cn (QC)

**Competing interest:** The authors declare that no competing interests exist.

**Abstract** Periodontitis drives irreversible destruction of periodontal tissue and is prone to exacerbating inflammatory disorders. Systemic immunomodulatory management continues to be an attractive approach in periodontal care, particularly within the context of 'predictive, preventive, and personalized' periodontics. The present study incorporated genetic proxies identified through genome-wide association studies for circulating immune cells and periodontitis into a comprehensive Mendelian randomization (MR) framework. Univariable MR, multivariable MR, subgroup analysis, reverse MR, and Bayesian model averaging (MR-BMA) were utilized to investigate the causal relationships. Furthermore, transcriptome-wide association study and colocalization analysis were deployed to pinpoint the underlying genes. Consequently, the MR study indicated a causal association between circulating neutrophils, natural killer T cells, plasmacytoid dendritic cells, and an elevated risk of periodontitis. MR-BMA analysis revealed that neutrophils were the primary contributors to periodontitis. The high-confidence genes *S100A9* and *S100A12*, located on 1q21.3, could potentially serve as immunomodulatory targets for neutrophil-mediated periodontitis. These findings hold promise for early diagnosis, risk assessment, targeted prevention, and personalized treatment of periodontitis. Considering the marginal association observed in our study, further research is required to comprehend the biological underpinnings and ascertain the clinical relevance thoroughly.

## eLife assessment

In this **fundamental** study, the authors analyzed associations between circulating immune cells and periodontitis. **Convincing** evidence identifies three immune cell types related to periodontitis, which substantially advances our understanding of periodontitis.

## Introduction

### Periodontitis imposes a considerable social burden on dental practice and general health

Periodontitis is a highly prevalent disease that affects a considerable percentage of the population. According to large-scale epidemiological research, up to half of all adults worldwide suffer periodontal disease, with severe periodontitis threatening 10.5–12.0% of them (*Kassebaum et al., 2014*). Furthermore, periodontitis is the leading cause of adult tooth loss, necessitating extensive dental procedures such as extractions, dental implants, or prosthetics, which can be costly and time-consuming for both patients and dental practitioners (*Genco and Sanz, 2020*). Recent research demonstrated a relationship between periodontitis and inflammatory comorbidities such as type 2 diabetes, cardiovascular disease, rheumatoid arthritis, and inflammatory bowel disease (*Hajishengallis and Chavakis, 2021*). The high prevalence and harmful implications of periodontitis underline the importance of managing periodontitis to maintain oral and general health (*Peres et al., 2019*). Since early-stage prevention is the most significant way to improve health, the identification of additional potential risk factors was required to provide predictive, preventive, and personalized strategies for periodontal care (*Ma et al., 2021*).

### Evidence from epidemiology and pathophysiology demonstrates the impact of circulating immune cells on periodontitis

Periodontitis is a chronic inflammatory disease characterized by the interactions between microorganisms and host immune response (*Curtis et al., 2020*). The immune response to periodontitis comprises both innate and adaptive immunity, with multiple cytokines, immune cells, and inflammatory pathways participating in a complex interplay (*Dutzan et al., 2016*). Systemic immunological alternations, such as circulating immune cells, play a crucial role in the initiation and progression of periodontitis (*Cekici et al., 2014*). An observational study indicated that patients with periodontitis experience a greater level of circulating leukocytes (*Noz et al., 2021*), while another discovered that the distribution of B cells alters in the context of severe periodontitis, with a higher proportion of circulating memory B cells (*Demoersman et al., 2018*). Furthermore, inflamed periodontal tissue recruits immune cells from circulation (*Hajishengallis, 2020*). With the progression of periodontitis, there is a significant alteration in the quantity of immune cells present within the periodontal tissue. Specifically, an increase in the count of both monocytes and B cells is observed, whereas a decrease is noted in the count of T cells (*Nair et al., 2014*; *Steinmetz et al., 2016*). The promising concept of 'trained immunity' has recently provided a greater understanding of the host immune response in periodontitis (*Netea et al., 2020*), which can explain the fact that the increased hyper-responsiveness of circulating immune cells from patients with periodontitis as well as its probable mechanism of mediating periodontitis and its comorbidities (*Li et al., 2023*).

### Immunomodulation of systemic immune response serves as a hub for periodontal care

Systemic immunomodulation management can potentially improve host homeostasis by altering the composition and function of the immune milieu (*Yang et al., 2021*). Periodontitis can be effectively managed by restricting immune cell activation, implying that immunomodulators have significant promise in constructing comprehensive strategies for periodontal management (*Zidar et al., 2021*). For example, resveratrol, quercetin, and *N*-acetylcysteine were reported to reduce the release of reactive oxygen species by neutrophils, which aided in the prevention of periodontitis (*Orihuela-Campos et al., 2015*). Nonetheless, from a medical and therapeutic perspective, it is critical to determine whether the link between circulating immune cells and periodontitis is merely correlative or driven by causative mechanistic interactions (*Lamont and Hajishengallis, 2015*). Understanding the role of systemic immune alternations in periodontitis is critical for developing an effective strategy for early screening of high-risk patients, prompt implementation of definitive prevention, and individualized deployment of targeted treatment, all to reduce unexpected inflammatory responses, maintaining oral health, and avoiding complications (*Zhang et al., 2023*).

### Mendelian randomization provides a potent complement to causal inference in terms of genetics

Previous research has substantiated the potential of immunomodulation management in predicting and preventing periodontitis; however, in observational studies, the association is frequently disguised

by reverse causality, confounding factors, and disease conditions, which obscured the intrinsic causal inference between them (*Hajishengallis and Korostoff, 2017*). Mendelian randomization (MR) investigates the causal relationships between risk factors and diseases by exploiting genetic variants as instrumental variables (IVs) (*Davies et al., 2018*), which is less likely to be affected by underlying bias or disease condition in that alleles are randomly allocated from parents to offspring (*Julian et al., 2023*). Notably, MR with distinct causal relationships may provide fresh evidence from a genomics perspective (*Golubnitschaja et al., 2014*). We postulate that individuals with a disproportionate immunological network have a higher risk of periodontitis due to unexpected inflammatory reactions.

## Results

The present study, as shown in *Figure 1*, was based on the Strengthening the Reporting of Observational Studies in Epidemiology using the Mendelian Randomization (STROBE-MR) checklist (*Skrivankova et al., 2021*). The present research aims to evaluate the causal association between circulating immune cells and the risk of periodontitis, providing insight into opportunity for systemic immunomodulation management in periodontal care. We utilized summary statistics from publicly available genome-wide association studies (GWASs) to perform both univariable MR (UVMR) and multivariable MR (MVMR) analyses (*Table 1*, *Supplementary file 1–Table S1*). Furthermore, we replicated the UVMR analysis by excluding potentially pleiotropic single-nucleotide polymorphisms (SNPs) and subsequently conducted subgroup and reverse MR analyses. Additionally, the Bayesian model averaging (MR-BMA) was employed to pinpoint the predominant characteristics with causal signals. Finally, we conducted a transcriptome-wide association study (TWAS) and colocalization analysis to discern potential genes implicated in biological relationships.

### Estimated effects of circulating immune cells on periodontitis risk

Following a rigorous screening procedure, a total of 1940 SNPs were selected as IVs in the present study (*Supplementary file 1–Table S2*). The $F$-statistics ranged from 28.67 to 220.07, indicating a low risk of weak instrument bias. Three circulating immune cells were identified to be suggestively significant in the inverse variance weighted (IVW) method [odds ratio (OR): 1.09, 95% confidence interval (CI): 1.01–1.17, p = 0.030 for natural killer T (NKT) cells; OR: 1.11, 95% CI: 1.00–1.23, p = 0.042 for neutrophils; OR: 1.13, 95% CI: 1.02–1.25, p = 0.025 for plasmacytoid dendritic cells (pDCs)], which were further supported by the maximum likelihood and MR Pleiotropy RESidual Sum and Outlier (MR-PRESSO) (*Figure 2A, B*; *Supplementary file 1–Table S3*). The MR-Egger regression revealed no evidence of horizontal pleiotropy (p-values for intercept >0.05). However, significant heterogeneity was detected in two traits (memory B cell and monocyte) (*Supplementary file 1–Table S4*), which faded after the removal of outliers (*Supplementary file 1–Table S5*—Table S5, *Figure 2—figure supplement 1*). Moreover, the leave-one-out analysis showed no influential SNPs significantly linked with the outcome (*Figure 2—figure supplement 2*). The observed significant results remained robust after removing pleiotropic SNPs (*Supplementary file 1–Table S6*), and the scatter plot displayed a balanced distribution among SNPs (*Figure 2C–E*).

A replication UVMR of three subgroups of periodontal diseases (chronic periodontitis, chronic gingivitis, and gingival hyperplasia) was also performed in the FinnGen cohort (*Figure 3*, *Figure 3—figure supplement 1*). However, the significant findings identified within the primary database were not replicated during subgroup analysis, which may be attributed to the heterogeneity of periodontal disease and variations in the population composition across datasets. Intriguingly, B cell was discovered to be involved in the subgroups of the FinnGen population (OR: 1.11, 95% CI: 1.02–1.22, p = 0.019 for chronic periodontitis; OR: 1.39, 95% CI: 1.02–1.88, p = 0.036 for gingival hyperplasia) (*Supplementary file 1–Tables S7–S9*). Reverse MR revealed no indication of reverse causality (*Supplementary file 1–Table S10*).

### Assessing the independent and prioritized relationships through MVMR

After accounting for variable mutual adjustment and covariate correction for potential confounders, the causal relationship between circulating neutrophils and periodontitis remained stable with no evidence of heterogeneity or pleiotropy (*Figure 4A, B*). Nevertheless, the observed association

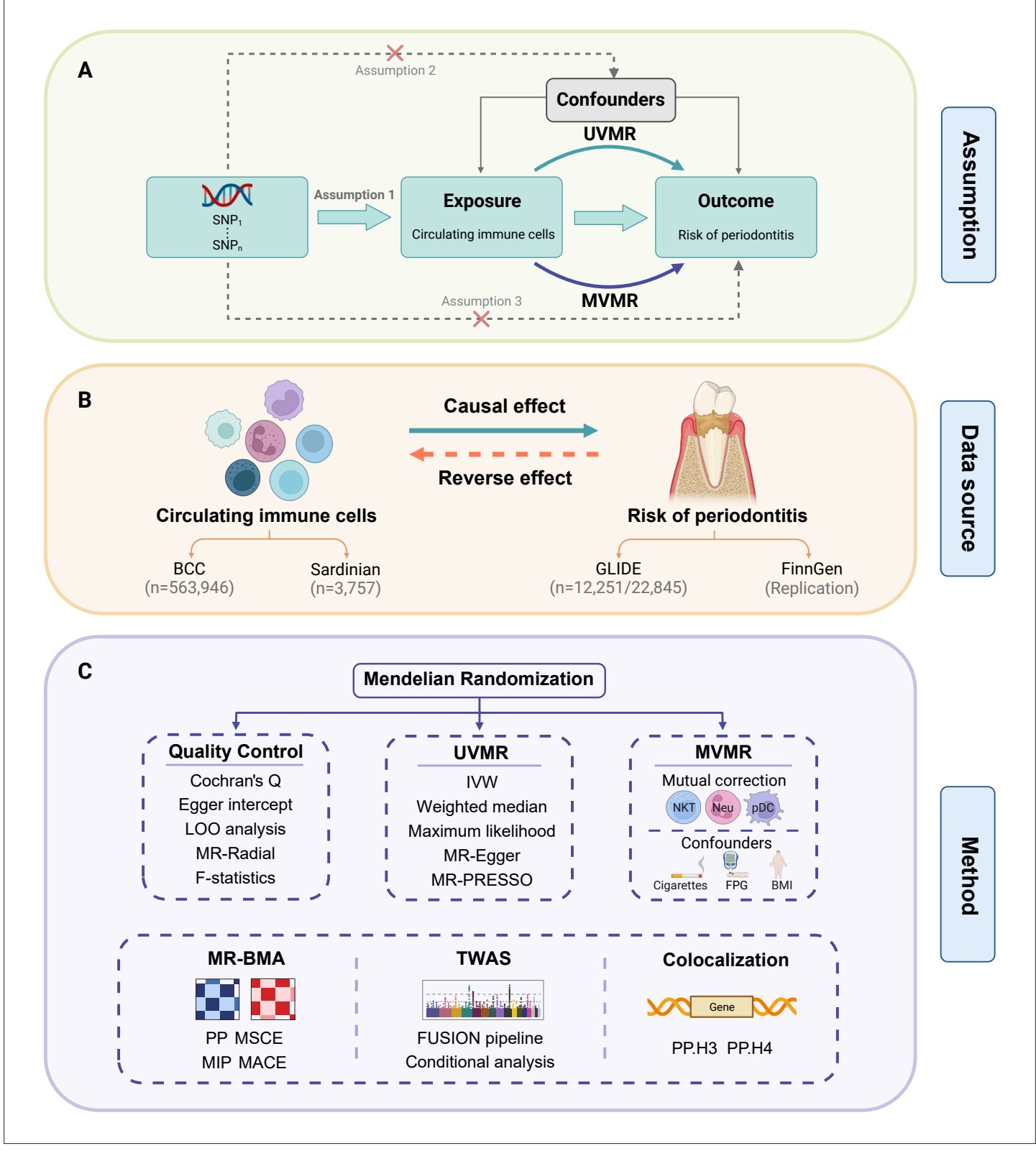

**Figure 1.** Study design. (**A**) Overview of the process and principal assumptions of MR. (**B**) Data sources of the GWASs. (**C**) Methods performed in the present study. Abbreviations and Notes: BCC, Blood Cell Consortium; BMA, Bayesian model averaging, a high-throughput method based on nonlinear regression; BMI, body mass index; FPG, fasting plasma glucose; FUSION, functional summary-based imputation; GLIDE, Gene-Lifestyle Interactions in Dental Endpoints collaboration consortium; GWAS, genome-wide association study; IVW, inverse variance weighted, the primary method in MR to

*Figure 1 continued on next page*

*Figure 1 continued*

explore the association between exposure and outcome; LOO, leave-one-out, a method for detecting potential influential SNPs; SNP, single-nucleotide polymorphism, as genetic instrumental variables for the exposure and outcome; MACE, model-averaged causal estimate; MIP, marginal probability of inclusion; MR, Mendelian randomization; MR-PRESSO, Mendelian Randomization Pleiotropy RESidual Sum and Outlier, a method for assessing and rectifying pleiotropic SNPs; MSCE, model-specific causal estimate; MVMR, multivariable Mendelian randomization, an MR model for adjusting confounding and mutual correction; Neu, neutrophil; NKT, natural killer T cell; pDC, plasmacytoid dendritic cell; PP, posterior probability; TWAS, transcriptome-wide association study; UVMR, univariable Mendelian randomization.

dissipated upon adjusting for body mass index (BMI), unveiling significant heterogeneity. Even though the MVMR-least absolute shrinkage and selection operator (MVMR-LASSO) analysis was utilized to make appropriate corrections, caution was still recommended when dealing with the effect. Furthermore, the significance of pDC and NKT remained stable after mutual adjustment, whereas the strength of the association for pDC was compromised in the MR-Egger sensitivity analysis (*Supplementary file 1–Table S11*).

In the MR-BMA framework, the best models and factors were ordered and prioritized based on their posterior probability (PP) and marginal inclusion probability (MIP) metrics (*Table 2*, *Supplementary file 1–Table S12*). Consequently, we observed neutrophil as the best model and leading factor for periodontitis (p = 0.771, MIP = 0.895), followed by NKT and pDC. The Cochran's Q test and Cook's distance failed to detect outlier or influential variations (*Figure 4C–E*).

## TWAS reveals key crosstalk genes

The TWAS indicated that five cross-trait genes, including *CC2D2B* (10q24.1), *RP11-326C3.7* (11p15.5), *USP3* (15q22.31), *HERC1* (15q22.31), and *AMFR* (16q13), may be implicated in the interaction of

**Table 1.** Characteristics of the GWAS data for MR.

| Phenotype | Year | Sample size (n case/n control) | n SNP (million) | Ancestry | Unit | Consortium/cohort | PMID |
|---|---|---|---|---|---|---|---|
| *Exposure* | | | | | | | |
| Circulating immune cells | 2020 | 563,946 | 15 | European | nl | BCC | 32888494 |
| Lymphocyte subsets | 2020 | 3757 | 15.2 | European | µg | Sardinian cohort | 32929287 |
| *Outcome* | | | | | | | |
| Periodontitis | 2019 | 35,096 (12,251/22,845) | 10.8 | European | Event | GLIDE | 31235808 |
| Chronic periodontitis (FinnGen) | 2023 | 263,668 (4434/259,234) | 20.2 | European | Event | FinnGen (R9K11) | |
| Chronic gingivitis (FinnGen) | 2021 | 196,245 (850/195,395) | 16.4 | European | Event | FinnGen (R5K11) | |
| Gingival hyperplasia (FinnGen) | 2023 | 259,613 (379/259,234) | 20.2 | European | Event | FinnGen (R9K11) | 36653562 |
| *Covariate* | | | | | | | |
| Cigarettes smoked per day | 2019 | 249,752 | 12 | European | 1/SD | GSCAN | 30643251 |
| Fasting plasma glucose | 2021 | 200,622 | 31 | European | mmol/l | MAGIC | 34059833 |
| Body mass index | 2018 | 681,275 | 2.3 | European | kg/m$^2$ | GIANT | 30124842 |

Abbreviations: BCC, Blood Cell Consortium; GIANT, GWAS of the Genetic Investigation of ANthropometric Traits; GLIDE, Gene-Lifestyle Interactions in Dental Endpoints; GSCAN, GWAS and Sequencing Consortium of Alcohol and Nicotine use; MAGIC, Meta-Analysis of Glucose and Insulin-related Traits Consortium; SD, standard deviation; SNP, single-nucleotide polymorphism; GWAS, genome-wide association study; MR, Mendelian randomization.

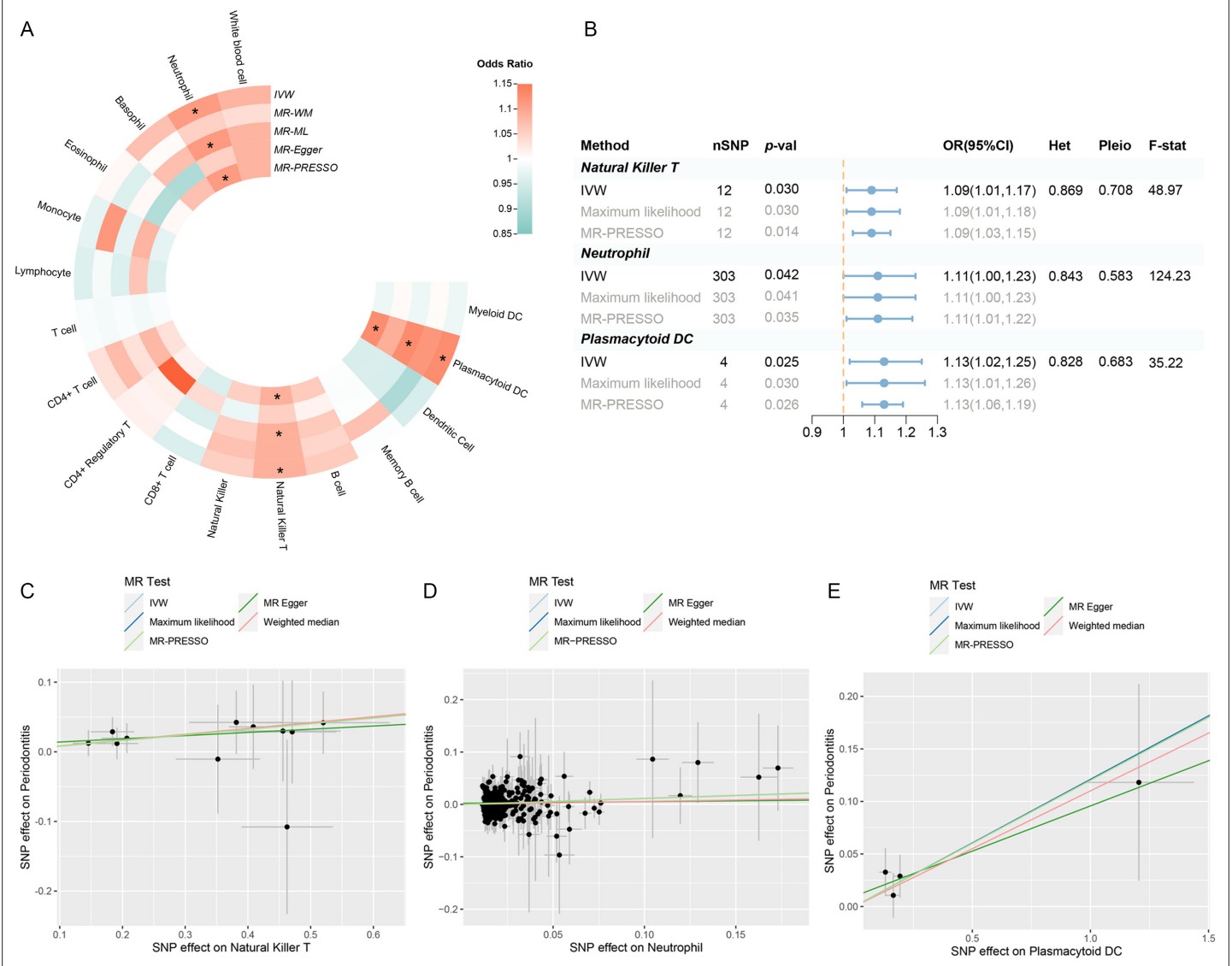

**Figure 2.** Results of the UVMR. (**A**) A circular heatmap representing the MR analyses for the associations between circulating immune cells and the risk of periodontitis. Lines, from outermost to innermost, represent IVW, MR-WM, MR-ML, MR-Egger, and MR-PRESSO, respectively. The color scale of the heatmap is based on the OR. *p < 0.05. (**B**) A forest plot of the MR analyses for significant results in (**A**) (p < 0.05). The effects are quantified using OR with 95% CI. (**C–E**) The effect estimates for each variant in natural killer T cell (**C**), neutrophil (**D**), and plasmacytoid DC (**E**) are provided by plotting SNP–outcome associations against SNP–exposure associations. Lines with different colors represent the regression slope fitted by different MR methods. Abbreviations: CI, confidence interval; DC, dendritic cell; *F*-stat, *F*-statistic; IVW, inverse variance weighted; Het, heterogeneity; MR, Mendelian randomization; MR-ML, Mendelian randomization weighted median; MR-WM, Mendelian randomization maximum likelihood; MR-PRESSO, Mendelian Randomization Pleiotropy RESidual Sum and Outlier; OR, odds ratio; Pleio, pleiotropy; SNP, single-nucleotide polymorphism; UVMR, univariable Mendelian randomization.

The online version of this article includes the following figure supplement(s) for figure 2:

**Figure supplement 1.** Scatter plots to explore outliers for two features with considerable heterogeneity.

**Figure supplement 2.** Results of leave-one-out sensitivity analysis.

circulating immune cells with periodontitis (***Figure 5A, B***). After Bonferroni correction (p < 6.27 × $10^{-6}$), we identified 658 of 3081 characteristics significantly associated with neutrophils, 5 of 443 with NKT, and 5 of 1038 with pDC. Within a broad criterion (p < 5 × $10^{-4}$), we discovered that 6 of 423 characteristics were linked to periodontitis (***Figure 5C, Figure 5—figure supplement 1***). Notably, four of these high-confidence genes were found to be involved with multiple phenotypes: *S100A9, S100A12* (neutrophils and periodontitis); *MCM6, P14KAP2* (neutrophils and pDC) (***Table 3, Figure 5D***). Most of

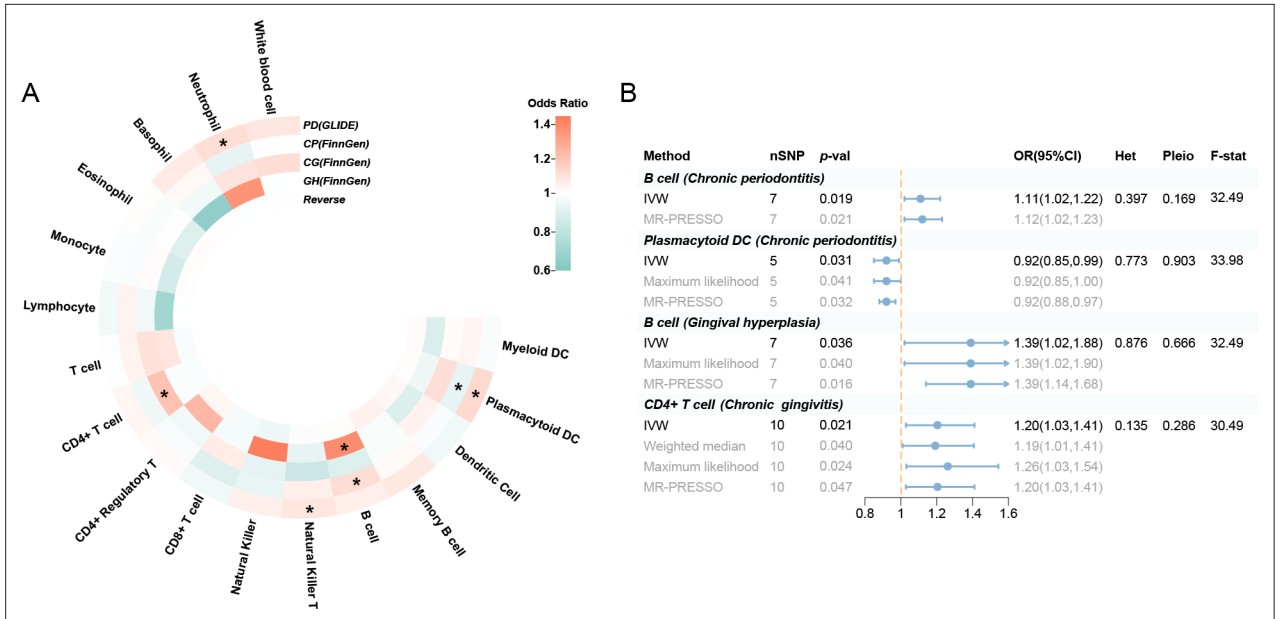

**Figure 3.** Results of the subgroup and reverse MR. (**A**) A circular heatmap illustrates the results of the subgroup analysis and reverse MR. Lines in the heatmap represent periodontitis (GLIDE), chronic periodontitis (FinnGen), chronic gingivitis (FinnGen), gingival hyperplasia (FinnGen), and reverse MR analysis, progressing from outside to inside. The color scale of the heatmap is determined by the odds ratio (OR). *p < 0.05. (**B**) A forest plot of the MR analyses for significant results in **Figure 4A** (p < 0.05). The effects are quantified using OR with 95% CI. Abbreviations: CG, chronic gingivitis; CI, confidence interval; CP, chronic periodontitis; DC, dendritic cell; *F*-stat, *F*-statistic; GH, gingival hyperplasia; GLIDE, Gene-Lifestyle Interactions in Dental Endpoints collaboration consortium; Het, heterogeneity; IVW, inverse variance weighted; MR-PRESSO, Mendelian Randomization Pleiotropy RESidual Sum and Outlier; PD, periodontitis; Pleio, pleiotropy; SNP, single-nucleotide polymorphism; MR, Mendelian randomization.

The online version of this article includes the following figure supplement(s) for figure 3:

**Figure supplement 1.** Scatter plots of the estimated effects in the subgroup MR.

---

these significant features survived conditional analysis and permutation testing (381/658 for neutrophils, 3/5 for NKT, 5/5 for pDC, and 5/6 for periodontitis). The majority of them were shown to be colocalized with their respective phenotype (554/658 for neutrophils, 4/5 for NKT, 3/5 for pDC, and 0/6 for periodontitis), implying that shared and pleiotropic SNPs influence both gene expression and phenotype (***Supplementary file 1–Tables S13–S16***).

## Discussion

In the present research, we employed MR to explore the potential links between circulating immune cells and periodontitis. Our study revealed causal relationships between elevated levels of circulating neutrophils, NKT cells, and pDCs with a higher risk of periodontitis. TWAS and colocalization analysis demonstrated possible high-confidence and cross-trait genes to be engaged in their interaction.

### Circulating neutrophils play a significant part in periodontitis and inflammatory comorbidities

Notably, our findings suggested that circulating neutrophils may play a leading causal role in the likelihood of periodontitis, and it remained robust after correcting for potential confounding factors and outliers. Neutrophils are abundant and short-lived myeloid cells that can be rapidly recruited to inflammatory sites, serving as the first line of defense against infections and other host insults. In recent years, a profound understanding of the role of neutrophils in chronic inflammatory diseases, where they may directly act as effectors of destructive inflammation, has been gained. However, a scarcity of neutrophils can also trigger damaging tissue inflammation, their pivotal role in maintaining physiological equilibrium (***Ley et al., 2018***). Numerous pieces of clinical evidence have uncovered that neutrophils account for a significant portion of inflammatory tissue damage and that the severity of periodontitis is positively correlated with the overproduction, dysregulation, or hyperactivity of

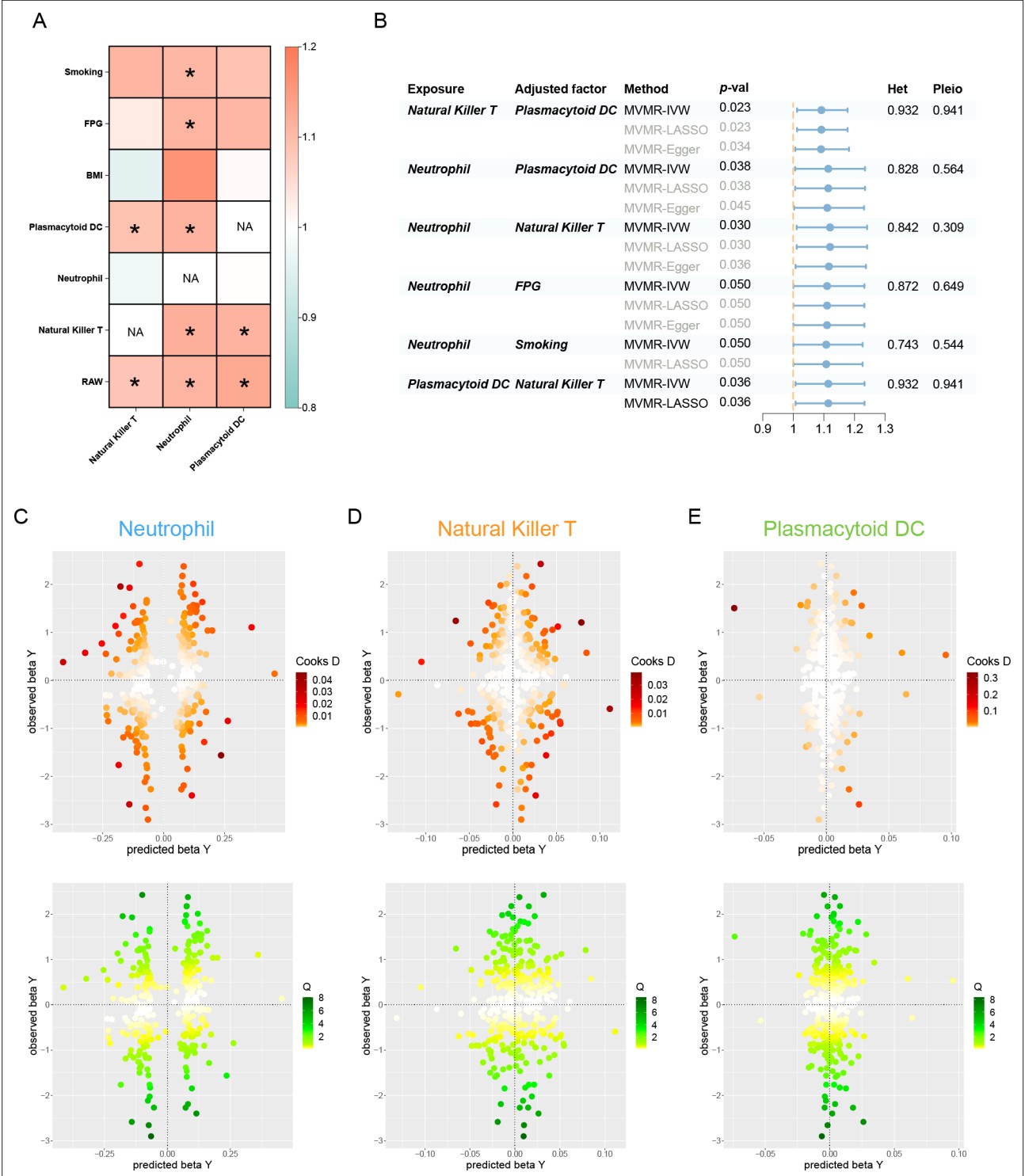

**Figure 4.** Results of the MVMR. (**A**) A heatmap represents the results of the MVMR. Rows and columns correspond to exposures and adjustment factors, respectively. The color scale of the heatmap is determined by the odds ratio (OR). *p < 0.05. (**B**) Forest plot analysis of the MVMR of significant results following mutual adjustment and confounder correction. The effects are quantified using OR with 95% CI. Scatter plots of the Cochran's Q test and Cook's distance to explore outlier or influential variations in Mendelian randomization-Bayesian model averaging (MR-BMA) for neutrophil (**C**), natural killer T (**D**), and plasmacytoid DC (**E**), respectively. Abbreviations: BMA, Bayesian model averaging; BMI, body mass index; CI, confidence interval; DC, dendritic cell; FPG, fasting plasma glucose; IVW, inverse variance weighted; Het, heterogeneity; LASSO, least absolute shrinkage and selection operator; MVMR, multivariable Mendelian randomization; Pleio, pleiotropy; SNP, single-nucleotide polymorphism.

**Table 2.** Ranking of risk factors and models for periodontitis in MR-BMA analysis.

| Trait | Ranking by MIP | MIP | MACE | Ranking by PP | PP | MSCE |
|---|---|---|---|---|---|---|
| Neutrophil | 1 | 0.895 | 0.097 | 1 | 0.771 | 0.108 |
| Natural killer T cell | 2 | 0.135 | −0.003 | 2 | 0.056 | −0.017 |
| Plasmacytoid DC | 3 | 0.102 | 0 | 3 | 0.045 | 0.007 |

DC, dendritic cell; MACE, average causal effect of risk factor model; MIP, marginal inclusion probability; MR-BMA, Mendelian randomization-Bayesian model averaging; MSCE, model-specific causal estimates; PP, poster probability.

neutrophils (*Chapple et al., 2023*; *Fine et al., 2021*). A case–control study indicated that periodontitis patients suffered from more apoptotic circulatory neutrophils than healthy people (*Nicu et al., 2018*). An increased neutrophil count could suggest the inflammatory burden of gingivitis and dental plaque in the oral cavity (*Sreenivasan and Prasad, 2022*). Another study discovered neutrophil depletion ameliorated experimental periodontitis, while unrestrained recruitment aggravated it (*Dutzan et al., 2018*).

Neutrophil-mediated inflammatory responses are crucial in the pathogenesis of periodontitis, influencing its localized manifestation and systemic complications, thereby associating it with a broad spectrum of inflammatory complications (*Hajishengallis and Chavakis, 2022*). During periodontitis, neutrophils act as the primary defense against bacteremia, with their primary function being to engulf and eradicate these bacteria, thus inhibiting their dissemination within the body. Following this, neutrophils liberally release proinflammatory cytokines in the connective tissue adjacent to the periodontal pocket. It is plausible that these cytokines could permeate into the bloodstream, thereby potentially exacerbating systemic inflammation (*D'Aiuto et al., 2013*). In severe periodontitis cases, patients experience low-grade systemic inflammation, as indicated by elevated levels of neutrophils and proinflammatory mediators, compared to their healthy counterparts (*Schenkein et al., 2020*). Preclinical research also demonstrates that ligature-induced periodontitis is accompanied by an elevation in circulating neutrophil counts, resulting in endothelial dysfunction and vascular inflammation (*Brito et al., 2013*). The quantity and functionality of neutrophils are critical indicators of inflammation severity. The reduction in neutrophil count and inflammatory mediators observed after successful periodontitis treatment suggests a decrease in systemic inflammation (*Hajishengallis et al., 2022*).

The recently developed concept of 'trained immunity' has introduced new perspectives on how neutrophils contribute to periodontitis and its comorbidities. 'Trained immunity' primarily denotes the memory-like response of myeloid innate immune cells, such as neutrophils and monocytes, to pathogens following initial infection (*Li et al., 2022*). Contrary to the memory response of specific immune cells, the memory characteristic of trained immunity primarily manifests as a rapid response and efficient elimination of pathogens rather than clear recognition (*Netea et al., 2016*). Trained myeloid cells have the potential to amplify the functionality of neutrophils, thereby fortifying the body's defense against subsequent infections. Nevertheless, within the framework of chronic inflammation, these cells could intensify tissue damage (*Hajishengallis, 2014*).

Periodontitis exemplifies a condition that stimulates maladaptive myelopoiesis in the bone marrow, characterized by producing excessive hyper-inflammatory monocytes and neutrophils (*Li et al., 2023*). This leads to an influx of maladaptively trained neutrophils in both the blood circulation and periodontal tissues that infiltrate not only oral tissues but also non-oral ones, simultaneously triggering an increase in the production of neutrophil extracellular traps and a reduction in their degradation (*White et al., 2016*). As a result, this process intensifies the collapse of the epithelial barrier, fosters bacteremia, exacerbates periodontitis, and amplifies the severity of inflammatory complications (*Burmeister et al., 2022*). While our comprehension of neutrophils' regulatory mechanisms and functions is far from complete, current research has facilitated the development of targeted therapeutic strategies to manage chronic inflammatory disorders mediated by neutrophils, such as the inflammatory response observed in periodontitis.

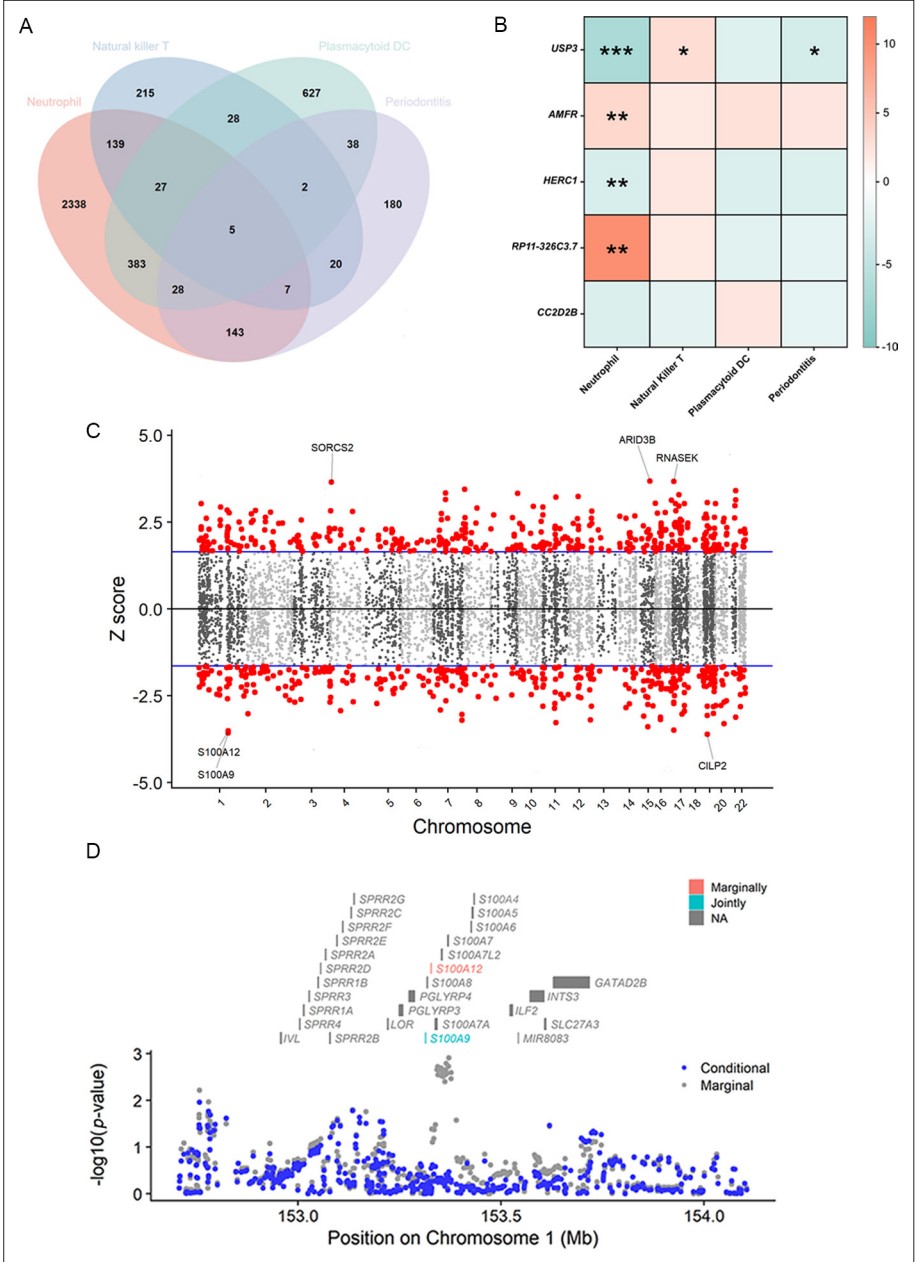

**Figure 5.** Results of the TWAS and colocalization analysis. (**A**) A Venn diagram illustrates the intersecting genes shared by multiple traits (p < 0.05). (**B**) A heatmap representing the TWAS and colocalization analysis for five genes interacting among neutrophil, natural killer T cell, plasmacytoid DC, and periodontitis. The TWAS Z-score is used as the color scale for the heatmap. *PP.H3 + PP.H4 > 0.5; **PP.H3 + PP.H4 > 0.8; ***PP.H4 > 0.8. (**C**) Manhattan plot of gene–traits associations for periodontitis. The x-axis represents genomic positions. Blue lines indicate a Z-score of 1.96. Red circles represent significant gene–trait associations (p < 0.05). Six genes satisfy a multiple corrected threshold of $p < 5 \times 10^{-4}$. (**D**) Regional Manhattan plot of conditional analysis for *S100A9*, *S100A12* in periodontitis. Gray bars indicate the location of genes on chromosome 1. Genes colored in orange and green on the graph indicate the marginally and jointly significant genes that best explain the GWAS signals. Gray and blue dots indicate the GWAS p-values before and after conditioning on the jointly significant gene. Abbreviations: DC, dendritic cell; GWAS, genome-wide association study; PP, posterior probability; TWAS, transcriptome-wide association study.

The online version of this article includes the following figure supplement(s) for figure 5:

**Figure supplement 1.** Manhattan plots for natural killer T and plasmacytoid DC.

**Table 3.** TWAS and colocalization analysis identified genes involved in multiple phenotypes.

| Gene name | Position | Phenotype | TWAS Z-score | TWAS p val | Perm p val | Model p val | PP.H3 | PP.H4 |
|---|---|---|---|---|---|---|---|---|
| | | Neutrophil | 7.225 | $5.0 \times 10^{-13}$ | 0 | $1.7 \times 10^{-17}$ | 0.035 | 0.962 |
| S100A9 | 1q21.3 | Periodontitis | −3.564 | $3.7 \times 10^{-4}$ | $1.3 \times 10^{-3}$ | $1.7 \times 10^{-17}$ | 0.008 | 0.259 |
| | | Neutrophil | 7.310 | $2.7 \times 10^{-13}$ | $1.7 \times 10^{-4}$ | $4.1 \times 10^{-43}$ | 0.027 | 0.970 |
| S100A12 | 1q21.3 | Periodontitis | −3.513 | $4.4 \times 10^{-4}$ | $8.1 \times 10^{-4}$ | $4.1 \times 10^{-43}$ | 0.008 | 0.244 |
| | | Neutrophil | −5.549 | $2.9 \times 10^{-8}$ | 0 | $3.5 \times 10^{-13}$ | 0 | 0.999 |
| MCM6 | 2q21.3 | Plasmacytoid DC | 7.172 | $7.4 \times 10^{-13}$ | $2.7 \times 10^{-3}$ | $3.5 \times 10^{-13}$ | 1.000 | 0 |
| | | Neutrophil | 5.345 | $9.0 \times 10^{-8}$ | $3.3 \times 10^{-3}$ | $3.7 \times 10^{-39}$ | 0.003 | 0.016 |
| PI4KAP2 | 22q11.21 | Plasmacytoid DC | 5.322 | $1.0 \times 10^{-7}$ | $3.2 \times 10^{-3}$ | $3.7 \times 10^{-39}$ | 0.009 | 0.045 |

DC, dendritic cell; Perm, permutation test; PP, posterior probability; TWAS, transcriptome-wide association study.

## Several lymphocyte subsets are causally associated with the risk of periodontitis

NKT cells, a distinct fraction of T lymphocytes, are linked to the pathophysiology of various inflammatory, osteolytic, and autoimmune diseases (*Godfrey et al., 2000*). Similar to our findings, previous research revealed more NKT recruited in periodontitis tissues (*Muthukuru, 2012*; *Yamazaki et al., 2001*). Several studies have demonstrated the tissue-specific function of NKT and highlighted its pathogenic role in periodontitis (*Aoki-Nonaka et al., 2014*; *Melgar-Rodríguez et al., 2021*), which may be attributed to the proinflammatory and immunoregulatory activities mediated by NKT, spanning from cytokine production to immune cell interactions (*Seidel et al., 2020*).

In addition, our study identified a convoluted causal relationship between pDC and periodontitis. Dendritic cells, as specialized antigen-presenting cells, play a crucial role in the modulation of the host immune response and may be related to bone loss during periodontitis (*El-Awady et al., 2022*; *Ginesin et al., 2023*). In response to viral encounters and infection, pDC represents a unique subgroup of DC that releases type I interferon (*Jego et al., 2003*). However, pDC is only discovered in a tiny percentage of healthy oral tissues, and more relevant clinical research still needs to be conducted (*Meghil and Cutler, 2020*; *Wilensky et al., 2014*). The involvement of pDC in periodontitis deserves further investigation.

## Systemic immunomodulation management for immune cells serves as a target for periodontal care

Periodontitis is a damaging inflammatory disease induced and exacerbated by the plaque biofilm and host immune response (*Moutsopoulos and Konkel, 2018*). The systemic immune response comprises both innate and adaptive immunity, with numerous cytokines, immune cells, and inflammatory pathways interacting in complex crosstalk during periodontitis, hinting that immunomodulation management may be an essential target for periodontal care (*Dutzan et al., 2016*; *Hajishengallis, 2014*).

Reactive periodontal therapies, which focus on plaque management, pocket depth reduction, and gingival bleeding eradication, only sometimes produce the intended results and fall short of a genuinely comprehensive approach to dental care (*Kornman et al., 2017*). Recently, a promising term 'P4 periodontics' (Predictive, Preventive, Personalized, and Participatory) has been introduced as a multilayer healthcare paradigm for the management of periodontitis, emphasizing the personalized responsiveness of treatment to disease (*Bartold and Ivanovski, 2022*). Modulating systemic host immune responses is particularly appropriate for predicting the progression and severity of periodontitis in persons whose periodontal condition is only slightly correlated with dental plaque (*Divaris et al., 2020*). MR contributes a novel approach to investigating systemic immunological alternations in periodontitis. A recent MR study evaluated the causal associations between circulating cytokines and the risk of periodontitis (*Huang et al., 2023*).

Our present study highlighted five genes (*USP3*, *AMFR*, *HERC1*, *CC2D2B*, and *RP11-326C3.7*) that may play a pivotal role in the communication between circulating neutrophils, pDC, NKT, and

periodontitis, as well as two high-confidence genes (*S100A9*, *S100A12*) situated within 1q21.3 as prospective gene targets for regulating circulating neutrophils during periodontitis. Our findings could pave the way for a novel preventive and therapeutic approach to modifying the systemic immunological equilibrium in periodontitis patients by modulating circulating immune cells. These findings enable the prediction of individuals at risk of periodontitis by screening specific immune imbalances, which could then be employed to prevent periodontitis and related inflammatory comorbidities, particularly in patients with systemic susceptibility factors.

## Strengths and limitations in the present study

The present study exhibited several strengths. First, under the premise of three key assumptions, MR is a powerful tool for explaining the relationship between complicated features (such as circulating immune cells) by successfully mitigating the effect of probable confounders and allowing for reasonable causal order. Second, a rigorous quality control process was conducted in accordance with the STROBE-MR checklist in multiple domains, including IVs selection, heterogeneity investigations, and removal of pleiotropic loci (*Supplementary file 2*—STROBE-MR Checklist). Third, we adopted a series of sensitivity tests and MVMR to rule out the impact of outlier, influential, or pleiotropic SNPs. Fourth, a novel method based on nonlinear Bayesian averaging was applied to explore the causal drivers of disease risk from a set of high-throughput risk factors. Finally, TWAS was used in conjunction with MR to identify achievable regulatory gene targets for periodontal care.

However, some limitations should be addressed when interpreting the results. To begin with, a need for GWAS databases hampered more comprehensive and precise analyses. As a result, we could not evaluate the impact of immune cells on distinct subsets of periodontal illnesses (such as gingival recession and periodontal abscess) or ethnic groups (such as East Asian and African). Second, the primary results from the IVW method were not stable across all alternative analyses, nor were they replicated within subgroups, implying that the findings had limited evidentiary power. Third, the considerable variation in sample size between the two exposure databases contributes to the discrepancies in the number of positive SNPs. Despite our exploration of multiple selection thresholds for IVs, the inconsistency in screening methods and the disparity in the included SNPs could potentially introduce bias. Fourth, none of the results satisfied the Bonferroni correction (p < 0.05/17 traits = 0.003), which may have inflated the rate of type I errors. Despite our best attempts to minimize potential confounding factors, interferences from unobserved pleiotropies could not be completely ruled out. Fifth, we relied on the whole-blood data for the FUSION algorithm due to the lack of transcriptome data associated with oral tissues (such as gums, oral mucosa, and alveolar bone) in the GTEx database. This has led to an excessive focus on systemic immunological changes, thereby overlooking the significance of alterations in local periodontal tissue immunity. Such an oversight could compromise the precision and pertinence of our research findings. Sixth, in addition to quantities, function abnormalities (such as dysregulation or hyperactivity) of circulating immune cells may also be related to the susceptibility and severity of periodontitis; however, our research failed to address this issue. Seventh, while most immune cells in the gingival crevicular fluid are derived from blood, the amount of circulating immune cells is influenced by more intricate factors, which may challenge the current causal inference. Finally, since MR evaluates causal inference from the standpoint of genetic variations, it may sometimes correspond differently to fact.

In conclusion, the present study provides suggestive evidence of the causal associations of genetically predicted circulating neutrophils, NKT cells, and pDCs on the risk of periodontitis, which shed light on the involvement of systemic immunological alterations in periodontitis etiology. Our findings may provide an innovative and evidence-based framework for systemic immunomodulation management in periodontal care, which can be valuable for early diagnostics, risk assessment, targeted prevention, and personalized management of periodontitis, especially for patients with systemic susceptibility factors. However, the effect estimation discovered in our study was marginal, prompting caution when transferring to clinical practice. More studies are required to comprehend our findings' biological plausibility and clinical applicability.

## Materials and methods

### Data source

Summary-level data on 17 circulating immune cells were obtained from large-scale GWAS conducted by the Blood Cell Consortium (BCC) and the Sardinian cohort (*Orrù et al., 2020* [GWAS Catalog: GCST0001391-GCST0002121]; *Vuckovic et al., 2020* [GWAS Catalog: GCST90002379-GCST90002407]). The GWAS data for periodontitis and its subtypes were supplied by the Gene-Lifestyle Interactions in Dental Endpoints collaboration (GLIDE) consortium and the FinnGen cohort (*Kurki et al., 2023*; *Shungin et al., 2019*). To maintain the homogeneity within the target group and minimize overlap, we performed a screening process on the population. Populations from the Latin American and UK Biobank were excluded from the periodontitis dataset (*Ye et al., 2023*). Characteristics of GWAS and included cohorts are highlighted in *Table 1* and *Supplementary file 1–Table S1*. The statistical analyses were performed using 'TwoSampleMR' (version 0.5.7)', 'MRPRESSO' (version 1.0), 'RadialMR' (version 1.1), 'mrbma' (version 0.1.0), and 'GagnonMR' (version 0.0.0.9) packages in R software (version 4.3.1).

Candidates for IVs underwent a thorough set of screening procedures. A complicated criterion was performed to equalize the sample disparities among databases. We initially filtered the p-values of the SNPs, followed by the selection of independent SNPs using the linkage disequilibrium approach. A rigorous threshold of $p < 1 \times 10^{-9}$ was applied to the database with an abundance of positive SNPs (as in the BCC consortium) to ensure the reliability of IVs. Otherwise, a relatively strict standard of $p < 1 \times 10^{-6}$ was initially adopted (as in the Sardinian cohort), and we would loosen it at $p < 5 \times 10^{-6}$ if less than three SNPs met this threshold (an essential requirement for MR-PRESSO analysis). The $R^2$ and $F$-statistics were introduced to demonstrate the degree of genetic variation explained and their relative impact on the outcomes, and SNPs with $F$-statistics <10 would be removed based on the first MR assumption (*Papadimitriou et al., 2020*). In addition, SNPs that exhibited a direct association with the outcome would also be deleted to support the third MR assumption. Palindromic and ambiguous SNPs were eliminated throughout the harmonization processes to ensure the reliability and validity of causal inference. In MVMR, we excluded SNPs in the major histocompatibility complex area (6p21.31) due to their complexity and confounding effects (*Burgess and Thompson, 2015*).

### Univariable Mendelian randomization

In UVMR, the IVW method was performed as the primary analysis, and four alternative MR methods, including weighted median, maximum likelihood, MR-Egger, and MR-PRESSO global test were employed for sensitivity testing to assess the robustness of the IVW estimates. The IVW assumes that all genetic variations meet the conditions and integrates calculations from multiple genetic variants by weighting them inversely to variances (*Sanderson et al., 2022*). The weighted median generates precise estimates when more than half of the SNPs are valid (*Gormley et al., 2023*). The maximum likelihood offers a normal bivariate distribution to estimate causal effects by maximizing the likelihood function with a linear relationship (*Xue et al., 2021*). MR-Egger provides estimates after accounting for possible horizontal pleiotropy discovered by its incorporated intercept test, albeit the estimates were frequently underpowered (*Bowden et al., 2016*). MR-PRESSO detects outliers that cause pleiotropy and generates estimates once these outlier SNPs are eliminated (*Verbanck et al., 2018*). The observed significant results were considered 'robust' if the effect of sensitivity analyses was identical to that of the IVW method, yielding a p-value <0.05.

### Multivariable Mendelian randomization

To gauge the individual influence of each variant, MVMR analysis with mutual adjustment was performed, followed by a correction for associated confounders (*Burgess and Thompson, 2015*). We have incorporated covariates, including the number of cigarettes smoked, fasting plasma glucose levels, and BMI into the MVMR analysis, given that these factors could indirectly affect systemic immune responses and inflammation (*Liu et al., 2023*). The MVMR-IVW method was utilized as the primary test, supplemented by the MVMR-LASSO and the MVMR-Egger method (*Bowden et al., 2016*). The MVMR-LASSO regression yields dependable estimations for moderate-to-high degrees of heterogeneity or pleiotropy, and it also assists in alleviating the potential effects of multicollinearity among variables (*Grant and Burgess, 2021*).

The heterogeneity and horizontal pleiotropy of the results were quantified using Cochran's $Q$-statistics and the intercept term in MR-Egger regression, respectively. The MR-Radial, a more sensitive method for outliers, would detect and remove outlier SNPs whenever heterogeneity or pleiotropy was discovered (*Bowden et al., 2018*). The leave-one-out analysis and scatter plot were conducted to detect influential SNPs.

## Bayesian model averaging

As a multivariate framework for high-throughput risk factors based on nonlinear regression, the MR-BMA was then employed to explore the leading traits responsible for outcome (*Zuber et al., 2020*). First, we used closed-form Bayes factors and independence priors to calculate each variant's PP and model-specific causal estimates. Next, the total PPs for all potential models were added to determine the MIP. The model-averaged causal estimate, which reflected the average direct effect of each metabolic trait on the outcomes, was also used to compare risk factors and interpret the directions. Finally, the best model was chosen, preferably based on ranking each model's MIP and PP values. The $Q$-statistic and Cook's distance were used to identify invalid outliers and influential variants within the model. A genetic variant was defined as either an outlier or an influential variant if it possessed a $q$-value exceeding 10 or its Cook's distance surpassed the median of the corresponding $F$-distribution (*Eledum, 2021*). The MR-BMA would be repeated once unqualified variations were discovered.

## Transcriptome-wide association study

We exploited the updated Genotype-Tissue Expression (GTEx) project Version 8 whole-blood data for TWAS analysis (*Gusev et al., 2016*). First, the functional summary-based imputation (FUSION) pipeline was used to infer the transcriptome associated with significant outcomes, among which the optimal gene expression model was chosen by comparing the values of $R^2$ provided by Bayesian sparse linear mixed models and multiple penalized linear regressions. A Bonferroni-corrected criterion of $p < 6.27 \times 10^{-6}$ (0.05/7890 genes) was adopted to measure statistical significance. Then, conditional analysis and permutation testing were implemented to assess the dependability and robustness of the gene transcript–trait relationships discovered through TWAS. Finally, we performed an expression quantitative trait locus colocalization analysis on TWAS-derived genes to determine whether the association was caused by a single causal SNP (PP.H4) or distinct causal SNPs (PP.H3). PP.H3 + PP.H4 > 0.8 was considered significant evidence of colocalization (*Wallace, 2020*).

## Acknowledgements

We would like to acknowledge all the GWASs for making the summary data publicly available, and we appreciate all the investigators and participants who contributed to those studies. We appreciate the BioRender's convenience in drawing *Figure 1* (https://www.biorender.com). Fundamental Research Funds for the Central Universities 2022FZZX01-33 Shan Wang. Fundamental Research Funds for the Central Universities 2023QZJH58 Zhiyong Wang. National Major Science and Technology Projects of China 81991500 and 81991502 Qianming Chen. Zhejiang University Global Partnership Fund 188170 and 194452307/004 Zhiyong Wang. Joint Natural Science Foundation of Zhejiang Province LHDMD23H300001 Shan Wang.

## Additional information

### Funding

| Funder | Grant reference number | Author |
| --- | --- | --- |
| Fundamental Research Funds for the Central Universities | 2023QZJH58 | Zhiyong Wang |
| National Major Science and Technology Projects of China | 81991500 | Qianming Chen |

| Funder | Grant reference number | Author |
|---|---|---|
| Zhejiang University Global Partnership Fund | 188170 | Zhiyong Wang |
| Joint Natural Science Foundation of Zhejiang Province | LHDMD23H300001 | Shan Wang |
| Fundamental Research Funds for the Central Universities | 2022FZZX01-33 | Shan Wang |
| National Major Science and Technology Projects of China | 81991502 | Qianming Chen |
| Zhejiang University Global Partnership Fund | 194452307/004 | Zhiyong Wang |

The funders had no role in study design, data collection, and interpretation, or the decision to submit the work for publication.

## Author contributions

Xinjian Ye, Conceptualization, Formal analysis, Visualization, Writing – original draft; Yijing Bai, Mengjun Li, Conceptualization, Writing – original draft; Yuhang Ye, Yitong Chen, Yuwei Dai, Conceptualization, Methodology; Bin Liu, Validation, Methodology; Shan Wang, Funding acquisition, Writing – review and editing; Weiyi Pan, Supervision, Writing – review and editing; Zhiyong Wang, Supervision, Funding acquisition, Project administration; Yingying Mao, Supervision, Project administration, Writing – review and editing; Qianming Chen, Supervision, Funding acquisition, Project administration, Writing – review and editing

## Author ORCIDs

Xinjian Ye ⓘ https://orcid.org/0000-0003-2300-6117
Yijing Bai ⓘ http://orcid.org/0000-0002-2917-0156
Yingying Mao ⓘ https://orcid.org/0000-0003-3644-9160
Qianming Chen ⓘ http://orcid.org/0000-0002-5371-4432

## Ethics

Since we utilized publicly available GWAS summary data or published studies, ethical committee approval was not required for this manuscript.

Reviewer #1 (Public Review): https://doi.org/10.7554/eLife.92895.3.sa1
Reviewer #2 (Public Review): https://doi.org/10.7554/eLife.92895.3.sa2
Author Response https://doi.org/10.7554/eLife.92895.3.sa3

# Additional files

## Supplementary files

• Supplementary file 1. Case definition and exclusion criteria in included genome-wide association studies (GWASs). Characteristics of the single-nucleotide polymorphisms (SNPs) used as instrumental variables (IVs) in the Mendelian randomization (MR). Effect estimates of causal associations between circulating immune cells and periodontitis risk. The heterogeneity and horizontal pleiotropy of the main results using Cochran's $Q$-statistics and MR-Egger intercept. Effect estimates of causal associations after excluding outliers in RadialMR for two features with considerable heterogeneity. Effect estimates of causal associations after excluding influential or outlier SNPs for significant results. Effect estimates of causal associations between circulating immune cells and chronic periodontitis, chronic gingivitis, and gingival hyperplasia in subgroup analysis. Effect estimates of the reverse associations between periodontitis and circulating immune cells. Effect estimates of causal associations after mutual and covariate correction in multivariable Mendelian randomization (MVMR). Ranking of models according to their posterior probability (PP) using Mendelian randomization-Bayesian model averaging (MR-BMA) analysis. Results of transcriptome-wide association study (TWAS), conditional analysis, permutation testing, and colocalization analysis on

neutrophil, natural killer T cell, plasmacytoid dendritic cell (DC), and periodontitis.

• Supplementary file 2. Strengthening the Reporting of Observational Studies in Epidemiology using the Mendelian Randomization (STROBE-MR) checklist.

• MDAR checklist

## Data availability

The data generated or analyzed during this study are available in this published article and its supplementary information files. The code and curated data for the present analyzed are available at GitHub (copy archived at *Code4Ye, 2024*).

The following previously published dataset was used:

| Author(s) | Year | Dataset title | Dataset URL | Database and Identifier |
|---|---|---|---|---|
| The GLIDE consortium | 2019 | GWAS summary statistics for dental caries and periodontitis | https://doi.org/ 10.5523/bris.2j2r qgzedxlq02oqbb4v mycnc2 | Data Repository for University of Bristol, 10.5523/ bris.2j2rqgzedxlq02oqbb4vmycnc2 |

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
