## [Editor Report · eLife assessment]

In this **fundamental** study, the authors analyzed associations between circulating immune cells and periodontitis. **Convincing** evidence identifies three immune cell types related to periodontitis, which substantially advances our understanding of periodontitis.

---

## [Referee Report · Reviewer #1 (Public Review)]

Ye et al. used Mendelian randomization method to evaluate the causative association between circulating immune cells and periodontitis, and finally screened out three risk immune cells related to periodontitis. Overall, this is an important and novel piece of work that has the potential to contribute to our understanding of the causal relationship between circulating immune cells related to periodontitis.

---

## [Referee Report · Reviewer #2 (Public Review)]

Summary:

This is a carefully done study containing interesting results.

Strengths:

These findings have significant implications for periodontal care and highlight the potential for systemic immunomodulation management on periodontitis, which is of interest to readers in the fields of periodontology, immunology, and epidemiology.

---

## [Author Response]

The following is the authors’ response to the original reviews.

**eLife assessment**
The authors analyzed the causative association between circulating immune cells and periodontitis, and reported three risk immune cells related to periodontitis. The significance of the findings is fundamental, which substantially advances our understanding of periodontitis. The strength of evidence is convincing.
**Reviewer #1 (Public Review):**
Ye et al. used Mendelian randomization method to evaluate the causative association between circulating immune cells and periodontitis and finally screened out three risk immune cells related to periodontitis. Overall, this is an important and novel piece of work that has the potential to contribute to our understanding of the causal relationship between circulating immune cells related to periodontitis. However, there are still some concerns that need to be addressed.

We sincerely appreciate the constructive feedback from the editor and reviewers, which has been instrumental in enhancing the quality of our manuscript.

(1) The authors used 1e-9 as the threshold to select effective instrumental variables (IVs), which should give the corresponding references. Meanwhile, the authors should test and discuss the potential impact of inconsistent thresholds for exposure (1e-9, 5e-6 were selected by the author respectively) and outcome IVs (5e-8) on the robustness of the results.

Thank you for your insightful comments. We have selected two GWAS databases as the data sources for the exposure group: the BCC Consortium with a sample size of 563,946, and the Sardinian cohort of 3,757. The considerable disparity in sample size between them may result in variations in outcomes, primarily showcased in the differences in positive SNP numbers. We, therefore, adopted an unconventional (non 5e-8) yet rigorously controlled screening strategy, an approach that is widely accepted in MR studies (Li et al., 2022; Liu et al., 2023). We believe that the present thresholds are sufficiently rigorous to guarantee the validity of the subsequent Mendelian randomization analysis.

However, employing two distinct methods in exposure screening is not typical, and we posit that this method can be viewed as an innovative strategy, providing a reference for future research dealing with two databases with significant discrepancies (Huang et al., 2023; Kong et al., 2023). As you perceptively noted, we acknowledge that this strategy may exert a certain influence on the research outcomes, and we have factored this potential limitation into our manuscript. “Third, the considerable variation in sample size between the two exposure databases contributes to the discrepancies in the number of positive SNPs. Despite our exploration of multiple selection thresholds for IVs, the inconsistency in screening methods and the discrepancy in the included SNPs could potentially introduce bias.” (Page 14)

As for the "outcome IVs with 5e-8" you mentioned, we didn't implement this screening threshold in the outcome IVs. Indeed, we applied the same screening criteria as specified at 5e-06 (refer to Stable 2). Is the statement that you're referring to the following: "Additionally, SNPs that displayed a direct association with the outcome would also be excluded to uphold the third MR assumption (P < 5e-8)" (Page 6)? In this context, we adopted a standard criterion in the IVs screening process to remove SNPs directly associated with the outcome.

Reference

Huang W, Wang Z, Zou C, Liu Y, Pan Y, Lu J, Zhou K, Jiao F, Zhong S, Jiang G. 2023. Effects of metabolic factors in mediating the relationship between Type 2 diabetes and depression in East Asian populations: A two-step, two-sample Mendelian randomization study. J Affect Disorders 335:120–128. doi:10.1016/j.jad.2023.04.114

Kong L, Ye C, Wang Y, Zheng J, Zhao Z, Li M, Xu Y, Lu J, Chen Y, Xu M, Wang W, Ning G, Bi Y, Wang T. 2023. Causal effect of lower birthweight on non-alcoholic fatty liver disease and mediating roles of insulin resistance and metabolites. Liver Int 43:829–839. doi:10.1111/liv.15532

Li P, Wang H, Guo L, Gou X, Chen G, Lin D, Fan D, Guo X, Liu Z. 2022. Association between gut microbiota and preeclampsia-eclampsia: a two-sample Mendelian randomization study. Bmc Med 20:443. doi:10.1186/s12916-022-02657-xLiuB, Lyu L, Zhou W, Song J, Ye D, Mao Y, Chen G-B, Sun X. 2023. Associations of the circulating levels of cytokines with risk of amyotrophic lateral sclerosis: a Mendelian randomization study. Bmc Med 21:39. doi:10.1186/s12916-023-02736-7 1

(2) What is the reference for selecting Smoking, Fasting plasma glucose, and BMI as covariates? They do not seem to be directly related to immune cells as confounding factors.

The variables of Smoking, Fasting Plasma Glucose (FPG), and Body Mass Index (BMI) are commonly used as covariates in multivariable Mendelian randomization studies (Kong et al., 2023; Liu et al., 2023). The association between Smoking, FPG, and BMI with immune cells may not be immediately apparent. However, these factors have been identified as potential confounders that could impact overall health, which in turn may indirectly modulate systemic immune responses, susceptibility, and inflammation.

(1) . Smoking: It has been well-documented that smoking can cause inflammation and impair immune function, thereby increasing individual's susceptibility to infections and diseases (Shiels et al., 2014). As such, smoking is recognized as a covariate that could potentially influence the outcomes of an investigation into immune cells.

(2) FPG: Elevated FPG levels indicate poor glycemic control, potentially leading to conditions like diabetes (Choi et al., 2018). Consequently, studies have demonstrated that elevated FPG levels can compromise the immune system's ability to combat infections.

(3) BMI: It is a measure of body fat that takes into account a person's weight and height. Both obesities, characterized by a high BMI, and underweights, characterized by a low BMI, have been associated with a range of health issues, inclusive of a compromised immune system (Piñeiro-Salvador et al., 2022). Consequently, BMI is factored in as a covariate in this study.

We have thus incorporated these factors as covariates in our study to mitigate their potential confounding effects. The selection of these covariates is primarily guided by previous research and established knowledge concerning the potential influences on immune function. We appreciate your query and will ensure to clarify this point in our revised manuscript. “We have incorporated covariates, including the number of cigarettes smoked, fasting plasma glucose (FPG) levels, and body mass index (BMI) into the MVMR analysis, given that these factors could indirectly affect systemic immune responses and inflammation (Liu et al., 2023).” (Page 6-7)

Reference

Choi S-C, Titov AA, Abboud G, Seay HR, Brusko TM, Roopenian DC, Salek-Ardakani S, Morel L. 2018. Inhibition of glucose metabolism selectively targets autoreactive follicular helper T cells. Nat Commun 9:4369. doi:10.1038/s41467-018-06686-0

Kong L, Ye C, Wang Y, Zheng J, Zhao Z, Li M, Xu Y, Lu J, Chen Y, Xu M, Wang W, Ning G, Bi Y, Wang T. 2023. Causal effect of lower birthweight on non-alcoholic fatty liver disease and mediating roles of insulin resistance and metabolites. Liver Int 43:829–839. doi:10.1111/liv.15532

Liu Y, Lai H, Zhang R, Xia L, Liu L. 2023. Causal relationship between gastro-esophageal reflux disease and risk of lung cancer: insights from multivariable Mendelian randomization and mediation analysis. Int J Epidemiol 52:1435–1447. doi:10.1093/ije/dyad090

Piñeiro-Salvador R, Vazquez-Garza E, Cruz-Cardenas JA, Licona-Cassani C, García-Rivas G, Moreno-Vásquez J, Alcorta-García MR, Lara-Diaz VJ, Brunck MEG. 2022. A cross-sectional study evidences regulations of leukocytes in the colostrum of mothers with obesity. BMC Med 20:388. doi:10.1186/s12916-022-02575-y

Shiels MS, Katki HA, Freedman ND, Purdue MP, Wentzensen N, Trabert B, Kitahara CM, Furr M, Li Y, Kemp TJ, Goedert JJ, Chang CM, Engels EA, Caporaso NE, Pinto LA, Hildesheim A, Chaturvedi AK. 2014. Cigarette smoking and variations in systemic immune and inflammation markers. J Natl Cancer Inst 106:dju294. doi:10.1093/jnci/dju294

(3) It is not entirely clear about the correction of P-value for the total number of independent statistical tests.

In our study, we used the Bonferroni correction to adjust the P-values for multiple comparisons. The adjusted P-value is calculated as the original P-value times the total number of independent statistical tests. Specifically, we applied multiple corrections in the following two aspects: First, we corrected the results of the FUSION algorithm in TWAS, with a correction value of P < 6.27 ×10-6 (0.05/7,890 genes) (Page 8). Second, we performed multiple corrections on the initial results of MR (P < 0.05/17 traits = 0.003). However, none of the results met the criteria after the correction, which is one of the limitations detailed in the discussion section of our study (Page 14).

(4) The author used whole blood data to apply FUSION algorithm. Although whole blood is a representative site, the authors should add FUSION testing of periodontally relevant tissues, such as oral mucosa.

We appreciate your insightful comments and suggestions. We concur that employing periodontally relevant tissues, like oral mucosa, for FUSION testing might yield more precise and pertinent results. However, in the Genotype-Tissue Expression project (GTEx) database, we could not find transcriptome data related to oral tissues, such as gums, oral mucosa, and alveolar bone (Review Table 1). Owing to the limitations of the database, in the context of our study, we primarily relied on whole blood data, given its availability and the extensive precedent documented in the literature for its utilization (Xu et al., 2023; Yuan et al., 2022).

We acknowledge that this is a limitation of our study and will certainly consider incorporating periodontally relevant tissues in our future research. In the revised manuscript, we have explicitly stated this limitation and underscored the necessity for additional studies to corroborate our findings with periodontally relevant tissues. Fifth, we relied on the whole blood data For FUSION algorithm due to the lack of transcriptome data associated with oral tissues (such as gums, oral mucosa, and alveolar bone) in the GTEx database. “Fifth, we relied on the whole blood data For FUSION algorithm due to the lack of transcriptome data associated with oral tissues (such as gums, oral mucosa, and alveolar bone) in the GTEx database. This has led to an excessive focus on systemic immunological changes, thereby overlooking the significance of alterations in local periodontal tissue immunity. Such an oversight could potentially compromise the precision and pertinence of our research findings.” (Page 15)

**Author response table 1. sa3table1:** Organizations and Samplesize in the GTEx database.

ID	Tissue	N	ID	Tissue	N
1	Muscle_Skeletal	706	28	Artery_Coronary	213
2	Whole_Blood	670	29	Brain_Cerebellum	209
3	Skin_Sun_Exposed_Lower_leg	605	30	Liver	208
4	Artery_Tibial	584	31	Brain_Cortex	205
5	Adipose_Subcutaneous	581	32	Brain_Nucleus_accumbens_basal_ganglia	202
6	Thyroid	574	33	Brain_Caudate_basal_ganglia	194
7	Nerve_Tibial	532	34	Brain_Cerebellar_Hemisphere	175
8	Skin_Not_Sun_Exposed_Suprapubic	517	35	Brain_Frontal_Cortex_BA9	175
9	Lung	515	36	Small_Intestine_Terminal_Ileum	174
10	Esophagus_Mucosa	497	37	Brain_Hypothalamus	170
11	Cells_Cultured_fibroblasts	483	38	Brain_Putamen_basal_ganglia	170
12	Adipose_Visceral_Omentum	469	39	Ovary	167
13	Esophagus_Muscularis	465	40	Brain_Hippocampus	165
14	Breast_Mammary_Tissue	396	41	Brain_Anterior_cingulate_cortex_BA24	147
15	Artery_Aorta	387	42	Cells_EBV-transformed_lymphocytes	147
16	Heart_Left_Ventricle	386	43	Minor_Salivary_Gland	144
17	Heart_Atrial_Appendage	372	44	Vagina	141
18	Colon_Transverse	368	45	Brain_Amygdala	129
19	Esophagus_Gastroesophageal_Junction	330	46	Uterus	129
20	Stomach	324	47	Brain_Spinal_cord_cervical_c-1	126
21	Testis	322	48	Brain_Substantia_nigra	114
22	Colon_Sigmoid	318	49	Kidney_Cortex	73
23	Pancreas	305	50	Bladder	21
24	Pituitary	237	51	Cervix_Endocervix	10
25	Adrenal_Gland	233	52	Cervix_Ectocervix	9
26	Spleen	227	53	Fallopian_Tube	8
27	Prostate	221	54	Kidney_Medulla	4

Reference

Xu J, Si H, Zeng Y, Wu Y, Zhang S, Shen B. 2023. Transcriptome-wide association study reveals candidate causal genes for lumbar spinal stenosis. Bone Joint Res 12:387–396. doi:10.1302/2046-3758.126.BJR-2022-0160.R1

Yuan J, Wang T, Wang L, Li P, Shen H, Mo Y, Zhang Q, Ni C. 2022. Transcriptome‐wide association study identifies PSMB9 as a susceptibility gene for coal workers’ pneumoconiosis. Environmental Toxicology 37:2103–2114. doi:10.1002/tox.23554

(5) The authors chose gingival hyperplasia as a secondary validation phenotype of periodontitis in this study. However, gingival recession, as another important phenotype associated with periodontitis, should also be tested and discussed.

We appreciate your insightful feedback highlighting the significance of incorporating gingival recession as a phenotype in periodontitis studies. Our emphasis on gingival hyperplasia in the study was primarily dictated by the initial study design and the data available from FinnGen R9K11. Notwithstanding the lack of gingival recession data in the available databases, we identified chronic gingivitis data in an earlier version of the Finnish database (FinnGen R5K11) as an alternative. We performed a Mendelian Randomization analysis on this dataset, with the results integrated into Supplementary Table 10. Concurrently, Table 1, Supplementary Table 1, Figure 4, and the corresponding descriptions in the manuscript were updated. We trust this adjustment can address the limitations identified in our research. We are confident that this not only augments the comprehensiveness of our study but also fosters a more holistic comprehension of periodontal disease.

(6) This study used GLIDE data as a replicated validation, but the results were inconsistent with FinnGen's dataset.

Thank you for your insightful comments and for bringing this issue to our attention. Indeed, it is of utmost importance to ensure the validity and reliability of our findings across various datasets. The observed inconsistency between the GLIDE data and FinnGen's dataset could be attributed to several reasons.

Firstly, this discrepancy might originate from the differences in population composition. The former is grounded on a comprehensive meta-analysis of cohorts focusing on periodontitis, whereas the latter utilizes a dataset from a full-phenotype cohort. In the former, the ratio of periodontitis to the control groups is approximately 1:2. In contrast, the ratio in the latter seems to be minuscule. The sample size in the FinnGen data may not suffice to detect the effects observed in the GLIDE dataset, given that larger exposure sizes enhance the ability to detect genuine associations.

Moreover, the heterogeneity of periodontitis can potentially result in variable outcomes. Phenotypic definition methods differ between the two databases. The GLIDE database diagnoses based on the criteria of Centers for Disease Control and Prevention/American Academy of Periodontology (CDC/AAP) and Community Periodontal Index (CPI) for physical signs. While the FinnGen database adopts the International Classification of Diseases (ICD) 10 standard for a comprehensive diagnosis. The former database employs a more practical yet broader standard for periodontitis, which might encompass pseudo-periodontitis.

Finally, the observed differences could be attributed to the variations in immune responses at distinct stages of periodontitis. During the initial stages of periodontitis, neutrophils and macrophages primarily mediate the immune response. With the progression of the disease, the involvement of T cells and B cells increases, thereby leading to a more intricate immune response (Darveau, 2010). Besides, the immune system's response to these oral health conditions is not uniform and can be influenced by multiple factors, including the individual's overall health, genetics, and lifestyle, potentially impacting the results (Hung et al., 2023).

Reference

Darveau RP. 2010. Periodontitis: a polymicrobial disruption of host homeostasis. Nat Rev Microbiol 8:481–490. doi:10.1038/nrmicro2337

Hung M, Kelly R, Mohajeri A, Reese L, Badawi S, Frost C, Sevathas T, Lipsky MS. 2023. Factors Associated with Periodontitis in Younger Individuals: A Scoping Review. J Clin Med 12:6442. doi:10.3390/jcm12206442

**Reviewer #2 (Public Review):**
This manuscript presents a well-designed study that combines multiple Mendelian randomization analyses to investigate the causal relationship between circulating immune cells and periodontitis. The main conclusions of the manuscript are appropriately supported by the statistics, and the methodologies used are comprehensive and rigorous.These findings have significant implications for periodontal care and highlight the potential for systemic immunomodulation management on periodontitis, which is of interest to readers in the fields of periodontology, immunology, and epidemiology.

We greatly appreciate the positive feedback and valuable insights provided by the reviewer, which have significantly contributed to the improvement of our manuscript.

**Reviewer #2 (Recommendations for The Authors):**
*AbstractLine 30-32: "Two-sample bidirectional univariable MR followed by sensitivity testing, multivariable MR, subgroup analysis, and the Bayesian model averaging (MR-BMA) were performed to explore the causal association between them. " What does the term "them" refer to here, please clarify it. The research method here is unclear, please reorganize it.Line 39: "S100A9 and S100A12" here should be italic.

We appreciate your meticulous suggestions and have revised the methods section accordingly. Additionally, the two genes have been highlighted in italics for emphasis.

"Univariable MR, multivariable MR, subgroup analysis, reverse MR, and Bayesian model averaging (MR-BMA) were utilized to investigate the causal relationships. Furthermore, transcriptome-wide association study (TWAS) and colocalization analysis were deployed to pinpoint the underlying genes." (Page 1)

IntroductionLine 78-80: "As reported, the number of immune cells in periodontal tissue changes as periodontitis progresses, featuring an increase in monocytes, and B cells and a decrease in T cells." Does the author mean that both monocytes and B cells increase as periodontitis progresses?

We are grateful for your meticulous reading and perceptive inquiries. We would like to confirm the accuracy of your understanding. In lines 78-80, our intended message was to communicate that with the progression of periodontitis, there is an increase in both monocytes and B cells in the periodontal tissue. This represents a typical immune response to the infection, where these cells play a pivotal role in counteracting periodontal pathogens. To enhance clarity, we have revised these lines in the manuscript as follows:

"With the progression of periodontitis, there is a significant alteration in the quantity of immune cells present within the periodontal tissue. Specifically, an increase in the count of both monocytes and B cells is observed, whereas a decrease is noted in the count of T cells." (Page 3)

MethodLine 164-165: "As the main test, the MVMR-IVW method, offered by the MVMR-least absolute shrinkage and selection operator (MVMR-LASSO), and the MVMR-Egger method were chosen." The author's expression here is ambiguous.

In response to your comment on the ambiguity in lines 164-165, we have revised the sentence for clarity. We hope this addresses your concern and clarifies our point more effectively.

"The MVMR-IVW method was utilized as the primary test, supplemented by the MVMR-least absolute shrinkage and selection operator (MVMR-LASSO) and the MVMR-Egger method." (Page 7)

Table 1: FinnGen has a greater sample size and more SNPs than GLIDE; why do authors choose the latter as the primary analysis?

Our choice to utilize GLIDE as the primary analysis tool, instead of FinnGen, was mainly guided by the specific research question we aimed to address. Despite FinnGen offering a larger sample size and more SNPs, GLIDE offers a more specialized and targeted dataset that suits the unique requirements of our study. In most MR studies, a similar strategy is adopted, wherein a large database of disease GWAS meta is utilized for exploration, followed by validation in full phenotype cohort (such as UKBiobank and FinnGen) (Liu et al., 2023; Yuan et al., 2023). To summarize, the reasons may primarily include the following:

Firstly, GLIDE offers a concentrated and targeted methodology for examining genetic data pertinent to periodontitis. This dataset is grounded in a comprehensive meta-analysis of cohorts centered on periodontitis, wherein the ratio of periodontitis cases to control groups is approximately 1:2. Conversely, the proportion in FinnGen seems to be negligible, given that it employs a dataset derived from a comprehensive phenotype cohort. Consequently, employing the GLIDE database as a primary investigative tool can generate more abundant genetic information associated with periodontitis.

Furthermore, the methodological facets of GLIDE align more accurately with the analytical framework of our study. For instance, the diagnostic criteria methods vary between the two databases. The GLIDE database derives its basis from the Centers for Disease Control and Prevention/American Academy of Periodontology (CDC/AAP) and Community Periodontal Index (CPI) for physical indicators. In contrast, the FinnGen database employs the International Classification of Diseases (ICD) 10 standard for an exhaustive diagnosis. The former adopts a more pragmatic, yet broader, standard for diagnosing periodontitis. The latter continues to use concepts of diseases such as "chronic periodontitis", which have been replaced by "periodontitis" in the latest disease classification from the "2017 World Workshop on the Classification of Periodontal and Peri-Implant Diseases and Conditions" in the periodontal field (Caton et al., 2018).

Reference

Caton JG, Armitage G, Berglundh T, Chapple ILC, Jepsen S, Kornman KS, Mealey BL, Papapanou PN, Sanz M, Tonetti MS. 2018. A new classification scheme for periodontal and peri-implant diseases and conditions - Introduction and key changes from the 1999 classification. J Clin Periodontol 45 Suppl 20:S1–S8. doi:10.1111/jcpe.12935

Liu Y, Lai H, Zhang R, Xia L, Liu L. 2023. Causal relationship between gastro-esophageal reflux disease and risk of lung cancer: insights from multivariable Mendelian randomization and mediation analysis. Int J Epidemiol 52:1435–1447. doi:10.1093/ije/dyad090

Yuan S, Xu F, Li X, Chen J, Zheng J, Mantzoros CS, Larsson SC. 2023. Plasma proteins and onset of type 2 diabetes and diabetic complications: Proteome-wide Mendelian randomization and colocalization analyses. Cell Rep Med 4:101174. doi:10.1016/j.xcrm.2023.101174

ResultLine 224: "The observed significant results remained robust after removing pleiotropic SNPs." It is not clear what the authors mean by "remain robust".Line 229-231: "The causal relationship between neutrophils and periodontitis remained stable with no evidence of heterogeneity or pleiotropy." It is also not clear what the authors mean by "remain stable". How does the author get to the conclusion that there is no evidence of heterogeneity or pleiotropy?Figure S5: Please offer a brief explanation on how to investigate outlier or influential changes using scatter plots and Cochran's Q test and Cook's distance.

Line 224: We apologize for the confusion caused by the term "remain robust". In the revised manuscript, we clarified this by stating, "The observed significant results are considered 'robust' if the effect of sensitivity analyses was identical to that of Inverse Variance Weighted (IVW) method, yielding a P-value less than 0.05." (Page 6)

Line 229-231: We used the terms "remain stable" and "remain robust" interchangeably to express the same idea. To clarify, we have now unified the expression in the revised manuscript. As for the conclusion of "no evidence of heterogeneity or pleiotropy", it is derived from the results of Cochran's Q and Egger's intercept tests (P<0.05). We have added this explanation to the revised manuscript for better clarity.

Figure S5: In the revised manuscript and Table, we have provided a succinct explanation regarding the investigation of outliers or influential changes as follows: " A genetic variant was defined as either an outlier or an influential variant if it possessed a q-value exceeding 10 or if its Cook's distance surpassed the median of the corresponding F-distribution. " (Page 7)

We have made all the necessary changes in the revised manuscript based on your comments. We hope our responses and revisions adequately address your concerns.

DiscussionI have consulted several pieces of literature to ensure a thorough explanation, which may be helpful for your writing.(1) Hajishengallis G, Li X, Divaris K, Chavakis T. Maladaptive trained immunity and clonal hematopoiesis as potential mechanistic links between periodontitis and inflammatory comorbidities. Periodontol 2000. 2022;89(1):215-230. doi:10.1111/prd.12421(2) Hajishengallis G, Chavakis T. Mechanisms and Therapeutic Modulation of Neutrophil-Mediated Inflammation. J Dent Res. 2022;101(13):1563-1571. doi:10.1177/00220345221107602

We appreciate your valuable feedback and the additional references you provided to enrich our manuscript. Upon receiving your comments, we have meticulously reviewed and incorporated the suggested literature into our revised manuscript. These references have furnished insightful information, which has been assimilated into the revised manuscript (Page 12) to enhance the explanation of the mechanisms of neutrophil-mediated inflammation and the potential association between periodontitis and inflammatory comorbidities.

"The quantity and functionality of neutrophils both act as critical indicators of inflammation severity. The reduction in neutrophil count and inflammatory mediators, observed after successful periodontitis treatment, suggests a reduction in systemic inflammation (Hajishengallis , 2022)." (Page 12)

"Trained myeloid cells have the potential to amplify the functionality of neutrophils, thereby fortifying the body's defense against subsequent infections. Nevertheless, within the framework of chronic inflammation, these cells could potentially intensify tissue damage (Hajishengallis and Chavakis, 2022)." (Page 12).